# Modelling the Control of Offline Processing with Reinforcement Learning

**Eleanor Spens[1]**
ellie.spens@ndcn.ox.ac.uk

**Neil Burgess[2]**
n.burgess@ucl.ac.uk

**Timothy Behrens[1,2]**
timothy.behrens@ndcn.ox.ac.uk

[1]University of Oxford    [2]University College London

## Abstract

Brains reorganise knowledge offline to improve future behaviour, with 'replay' involved in consolidating memories, abstracting patterns from experience, and simulating new scenarios. However, there are few models of how the brain might orchestrate these processes, and of when different types of replay might be useful. Here we propose a framework in which a meta-controller learns to coordinate offline learning of a lower-level agent or model in 'sleep' phases to maximise reward in a 'wake' phase. The meta-controller selects among several actions, such as learning from recent memories in a hippocampal store, abstracting patterns from memories into a 'world model', and learning from generated data. In addition, the meta-controller learns to estimate the value of each episode, enabling the prioritisation of past events in memory replay, or of new simulations in generative replay. Using image classification, maze solving, and relational inference tasks, we show that the meta-controller learns an adaptive curriculum for offline learning. This lays the groundwork for normative predictions about replay in a range of experimental neuroscience tasks.

## 1 Introduction

Brains are busy during rest, reorganising knowledge to improve future behaviour. In particular, rest improves the ability to generalise, infer 'missing links', and learn statistical patterns [1, 2]. This is linked to replay, the hippocampal reactivation of experiences in temporally compressed form [3]. Offline replay has several proposed functions: Firstly, it is thought to reactivate real memories to consolidate them into neocortex [4, 3] or to abstract general patterns from episodes [5]. Secondly, replay can generate new experiences [6], sometimes referred to as generative replay, which may help generalisation [7] and lessen catastrophic forgetting [8]. In addition, replay is thought to recombine conceptual 'building blocks' to extrapolate beyond direct observation [7, 9]. Excitingly, recent studies show that the content of replay depends on the stage of learning [10] and the stage of cognitive development [11], suggesting variation in the brain's 'strategy' for offline learning over both short and long timescales. (See Appendix A.2 for additional neuroscience context.)

One might expect the optimal balance of these offline processes to vary based on task demands, the uncertainty of the environment, and the reliability of internal models. Machine learning involves similar trade-offs: collecting real-world data may be expensive, while synthetic data may not be reliable. Whilst 'data augmentation' is often used to expand datasets by specifying label-invariant transformations (e.g. adding noise, rotating, or cropping an image in the case of image classification), identifying the right transformations can be a process of trial and error. As a result, training data tends

39th Conference on Neural Information Processing Systems (NeurIPS 2025).

to be curated and ordered based on human judgement. The fields of meta reinforcement learning [12], curriculum learning [13] and data valuation [14, 15] explore solutions to some of these issues.

Drawing together these ideas, we propose a framework in which a meta-controller learns to select between different offline actions in a 'sleep' state to maximise reward in a 'wake' state. In the simulations of image classification and maze solving, these actions are i) learning from recent memories in a hippocampal store, ii) abstracting patterns from memories into a 'world model', and iii) learning from data generated with this 'world model'. In the case of relational inference, the model has a more complex range of actions, including the abstraction of relational patterns, and the application of those patterns to infer new relationships. This framework could be used to make normative predictions about the types of offline processing and content of replay observed in neuroscience data. It also suggests how access to 'metacognitive actions', with which an agent can process its own knowledge offline, can improve the performance of a reinforcement learning agent, as can the ability to construct a custom curriculum for this learning.

## 1.1 Previous work

Reinforcement learning (RL), which has long used experience replay to improve performance, is a common framework in which to consider offline learning. In particular, Dyna [16] is an approach that uses real experiences to update both a policy and a learned model of the environment, which is then used to generate simulated experiences for additional updates, improving the sample efficiency of learning. However, Dyna and subsequent algorithms [17] rely on fixed heuristics to schedule real-world and synthetic updates. Meta RL refers to approaches in which a higher-level agent guides the learning of a lower-level agent [12]. Our work extends these ideas to metacognitive control over offline processes in a neuroscience context, and is related to the suggestion that the prefrontal cortex functions as a meta-controller [18].

Many studies in the RL literature explore ways to prioritise experience replay, either at the level of episodes or individual transitions [19], e.g. based on reward, uncertainty, or prediction error [20]. This often requires balancing selective replay with random sampling to maintain some diversity in the training data. Other studies suggest different prioritisation criteria for experience replay at different stages of learning [21]. Meanwhile neuroscience research suggests that memories are replayed in order of utility, i.e. the amount they would improve decision-making [22], or based on their longer-term value for 'map-building' [23]. Note that methods for prioritising experiences are not only useful for replaying past data – they can also guide which new experiences an agent should gather next in online learning, e.g. to enable efficient exploration [24].

Curriculum learning refers to the arrangement of training examples into an order that benefits learning, e.g. in which the model is exposed to data of an appropriate difficulty level at each stage [13]. Meanwhile data valuation is a growing field of applied machine learning which explores how to select the most useful samples with which to train a model, e.g. by estimating the marginal contribution of a given example [15], or by training a network to select the best examples via RL [14]. The data valuation aspect of our framework can be thought of in terms of learning a curriculum for offline learning.

## 2 The model

In each simulation, we consider alternating sleep and wake states (Figure 1a). During the wake state, the meta-controller receives new observations (images, maze navigation episodes, or observed relationships) from the environment which are stored in its model hippocampus. In addition, a reward is obtained reflecting performance on some task (image classification, maze navigation, or relational inference). During the sleep state, the meta-controller takes actions which correspond to different ways to process offline knowledge, gradually learning to act offline in a way that maximises the awake reward.

More formally, consider a meta-controller which co-ordinates the offline learning of a lower-level RL agent. For each sleep/wake cycle $n = 1, \ldots, N$:

1. **Sleep phase:** A meta-controller takes $T$ offline actions $a_{n,0:T-1}$ using its recurrent proximal policy optimisation (PPO; [25]) policy $\pi_\phi(a \mid x)$, where x is an observation capturing the state of

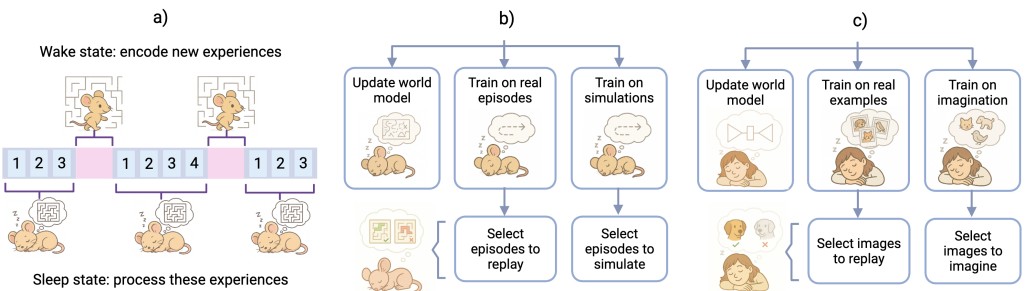

Figure 1: The model. a) In all simulations, a meta-controller learns to select from a range of possible offline actions during sleep, receiving a reward only at the end of the episode based on performance in the wake state. b) The maze task involves learning to navigate to a goal in a gradually changing maze. The meta-controller chooses between training a lower-level agent on real episodes from the hippocampus, updating its 'world model' of the maze's transition statistics from the hippocampus, and training the lower-level agent on episodes simulated with the world model. In addition, the system learns which episodes are most valuable to replay or to simulate. c) In the image task, images are encoded in the hippocampus during the wake state, where the distribution of classes varies, and the goal is to learn to classify them accurately. The meta-controller chooses between training a classifier on real images from the hippocampus, updating a simple generative (Gaussian mixture) model on images from the hippocampus, and training a classifier on generated images. In addition, the system learns which images are most valuable to replay or to generate.

the system. These actions involve offline processing of current memories and knowledge, *without* interacting with the real environment.

2. **Wake phase:** The agent is tested on the task for $n_{\text{test}}$ episodes, recording the evaluation reward $R_n = \frac{1}{n_{\text{test}}} \sum_{m=1}^{n_{\text{test}}} r_m$ where $r_m$ is the total reward in episode $m$. The meta-controller's policy is updated based on this reward.

   The agent interacts with the environment - which may have changed since the previous wake phase - for $K$ steps, encoding memories $H_n = \{(s_{k-1}, a_{k-1}, s_k, r_k)\}_{k=1}^{K}$ in the model hippocampus (here simply a list).

The lower-level task can be an arbitrary supervised learning task rather than an RL one by replacing the references to episodes above with observations and targets, and storing memories as $H_n = \{(\text{observation}_i, \text{target}_i)\}_{i=1}^{K}$.

The system also learns which items are most valuable to replay or generate, as follows. Truth data for the value is obtained for a small subset of candidate items by calculating their utility with a Shapley value approach, which estimates the change in performance if an example were omitted [15]. Then a small model is trained on this subset to predict the value of items (with the model fine-tuned each time data is selected rather than trained from scratch so that value estimates improve over time). Crucially, the data valuation procedure estimates the value of an example for subsequent learning *contingent on the state of the current model*, allowing it to capture the value of examples at different stages of training; this differs from some existing approaches which compute Shapley values with respect to a surrogate model trained from scratch, e.g. [26]. Given predicted values for *all* candidates, items are selected using maximal marginal relevance (MMR; [27]), which balances selecting the most valuable examples with promoting variety. Note that data valuation can be used to either undersample or oversample from a set of candidates. The data valuation component can be trained at the same time as the meta-controller, or separately for efficiency as shown here, in which case random sampling is used during meta-controller training. Whilst the meta-controller is frozen during inference, the value estimator network continues to be updated each time memory or generative replay is selected.

Our experiments used Stable-Baselines3 [28] for the RL algorithms, Gymnasium [29] to create the custom environments, and PyTorch [30] for other neural network training. See Appendix A.1 for further details.

# 3 Results

## 3.1 Image task

At the start of each episode 200 new images are stored in a buffer representing the hippocampus. The reward in the wake state, after three offline steps in the asleep state, is the accuracy of a classifier. Fashion-MNIST (a more challenging variant of MNIST featuring images of ten items of clothing) is used as a toy dataset [31].

The meta-controller agent is a recurrent PPO agent, enabling it to learn an optimal sequence of offline actions. The agent's inputs are the number of steps since the sleep phase began, the fraction of observations per class in the buffer, and the most recent action. The action space consists of the following actions (see Figure 1c):

- **Action 0**: Update the image generator with stored images from the model hippocampus. Specifically, the image generator is a Gaussian mixture model from which samples can be drawn for a particular class.
- **Action 1**: Train the classifier on *stored* images from the model hippocampus, selecting the 50 most useful 'memories' to replay with the value estimator model.
- **Action 2**: Train the classifier on *generated* images. As above, the 50 most useful items for 'generative replay' are selected using the value estimator model.

Note that we assume a small validation set exists for the data valuation element. To describe this more concretely, if the hippocampus contained many images, the marginal contribution of each of a subset (here, 25%) of images to the classification accuracy on a small validation set would be obtained. Then the value estimator (a linear regression model) would learn to estimate these values. Predicting the value of the many candidate items could then be performed efficiently with the value estimator, with these estimates used to prioritise which items to replay. (But in principle this validation set could be generated by the generative model, as we demonstrate for the maze task.)

### 3.1.1 Image results

Catastrophic forgetting or interference occurs when newly learned information overwrites previously learned information in a neural network, leading to poor performance on the previously learned tasks [32]. Continual learning is the ability to learn a series of tasks sequentially without the occurrence of catastrophic forgetting [33], and poses a general challenge for connectionist models of the brain. The training data for machine learning problems is carefully curated, e.g. to balance the number of examples of different categories, and shuffled before training. Reality is far messier, so biological learning must be robust to changing distributions over time.

To explore whether the agent can learn a strategy for continual learning, we compare a balanced and an imbalanced experience case. In the balanced experience case, examples from all ten Fashion-MNIST classes are encoded in the hippocampus in each wake state, and the model's accuracy on all ten classes is returned as feedback. In the imbalanced experience case, examples from two classes are encoded in the hippocampus in each wake state, and the model's accuracy on all ten classes is returned as feedback. This is sometimes referred to as class-incremental learning in the continual learning literature [34]. (Note that the image classification, generation, and valuation models are re-initialised when all categories of the dataset have been encoded, i.e. every five episodes in the imbalanced case.) We find that in the balanced experience case, replaying from the hippocampus is optimal (Figure 2b), but in the imbalanced experience case, training the generative model then learning from simulations is (Figure 2a). Unlike the baselines in Table 1, the meta-controller approach can do well in both settings, using generative replay to reduce catastrophic forgetting where necessary. (Note that the Gaussian mixture model learns a distribution per class, so in generative replay an equal number of samples are taken per class, from all classes added to the model so far.)

The behaviour of the data valuation component of the model was explored with i) the original Fashion-MNIST dataset (Figure 2d and g), and ii) a noisy version of the same dataset in which a randomly chosen fraction of pixels (25%, 50%, 75% or 100%) was set to zero in each image (Figure 2c and f), intended as a simple check that data valuation improved performance. Both variants were tested in the class-balanced case, hard-coding the actions to three stages of either memory replay or generative replay. That is, we enforced three consecutive actions and recorded the estimated values

of each image at each stage of memory or generative replay. Figure 2c and f show that selecting the images based on value estimates plus MMR results in significantly better performance than random selection, for memory replay and generative replay respectively. Table 2 shows it performs better than several other baselines too.

Figure 2d and g show that novelty (the distance from the mean of the class) is negatively correlated with estimated value in the first step of learning, but positively correlated by the third, for both memory replay and generative replay. One way to interpret this is that novel examples are *less* useful at first because they are too challenging, as observed in curriculum learning [13], but become *more* useful over time. We visualise value in the UMAP projected datasets in Figure 2i, and also corroborate this finding with MNIST (Figure 2e and h).

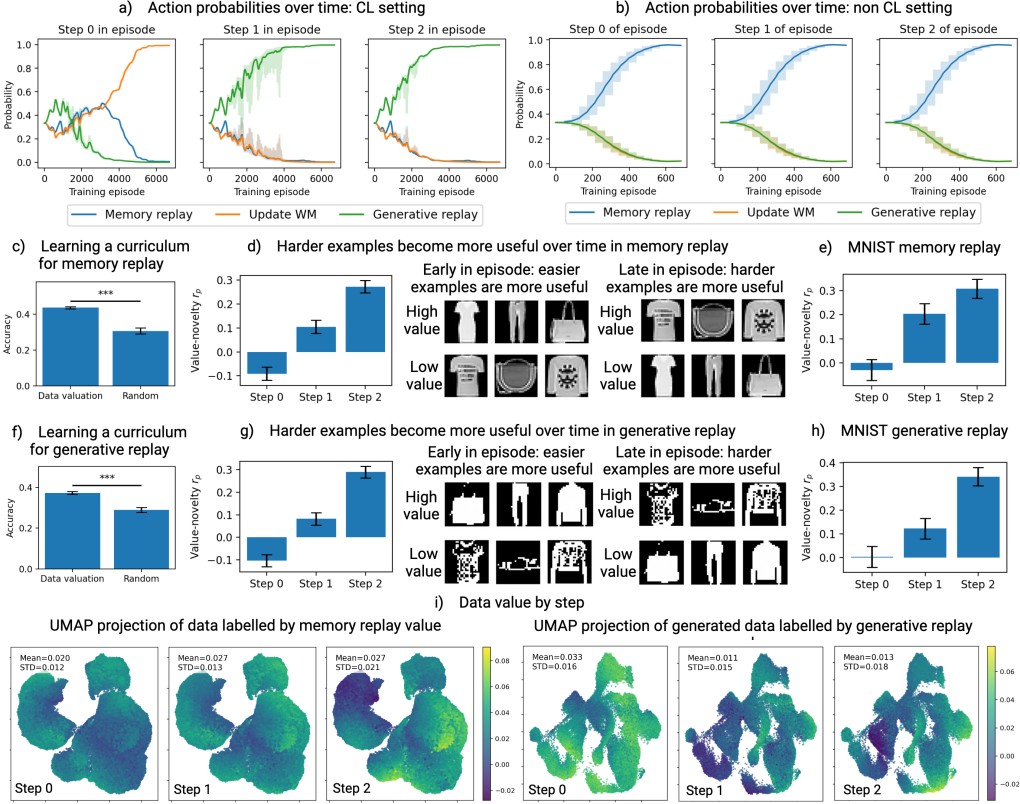

Figure 2: Image simulations. a) Action probabilities for each position in the sleep sequence over the course of training in the continual learning (CL) setting, in which the distribution of the agent's experiences is imbalanced, with only two classes observed in each wake phase. Shading indicates the SD across a sliding window of 100 episodes. (In other words, step 0 is the first action chosen in the sleep phase, step 1 is the second, and so on.) b) Action probabilities for each position in the sleep sequence over the course of training, but in the balanced experience baseline case. c) Learning to select the images for memory replay results in significantly better performance than random selection. The means across 10 repetitions are shown, and error bars give the SEM. Fashion-MNIST is used with the addition of random noise (setting a variable fraction of pixels to zero), with a balanced class distribution (i.e. the non-CL setting above). d) Here we take the original Fashion-MNIST dataset (no noise), and look at the relationship between novelty (distance from the class mean) and value for three consecutive stages of memory replay. At first the correlation is negative, indicating that easier examples are more valuable. But by stage three, the correlation is positive, indicating that harder examples are more valuable. e) Repeating the experiment in part d but for the MNIST dataset displays a similar pattern. f-h) As in parts c-e but for generative replay. Note that three steps of generative replay, following the training of the 'world model', are shown for consistency with the memory replay results. i) A projection into 2D with UMAP of the Fashion-MNIST data and their value estimates for memory replay (left), and of the generated data and their value estimates for generative replay (right).

Table 1: Image task results by approach. The meta-controller learns to use generative replay in the continual learning (CL) setting but memory replay in the non-CL setting. Results are averaged across 25 meta-episodes in each setting.

| Approach | Acc. (CL) | Acc. (non-CL) | Mean acc. | SEM |
|---|---|---|---|---|
| Meta-controller | **0.307** | **0.411** | **0.359** | 0.010 |
| Memory replay only | 0.171 | **0.411** | 0.291 | 0.003 |
| Generative replay only | **0.307** | 0.355 | 0.331 | 0.011 |

Table 2: Data valuation network for the image task compared to other data selection baselines, for choosing data to replay from the model hippocampus (memory replay; MR) or simulate (generative replay; GR). Ten meta-episodes are tested in each case. Diverse random selection uses the MMR method but with random data values. For the most and least challenging image baselines, the difficulty of an image is quantified as the fraction of an ensemble of scikit-learn classifiers that labelled the image correctly.

| Approach | Mean acc. (MR) | SEM (MR) | Mean acc. (GR) | SEM (GR) |
|---|---|---|---|---|
| Data valuation network | **0.438** | 0.007 | **0.373** | 0.007 |
| Random selection | 0.307 | 0.017 | 0.290 | 0.013 |
| Diverse random selection | 0.378 | 0.012 | 0.273 | 0.018 |
| Most challenging | 0.145 | 0.014 | 0.300 | 0.009 |
| Least challenging | 0.364 | 0.009 | 0.300 | 0.009 |

### 3.2 Maze task

Next we model a maze solving task in which the maze gradually changes over time, based on a rodent neuroscience task [35]. Here the higher-level agent is a recurrent PPO agent as above, but the lower-level agent for navigating from a random start to a random reward location in the environment is a Deep Q-Network (DQN; [36]) agent. The inputs to the DQN agent are one-hot encoded representations of the current and goal locations. The maze is simply a six-by-six grid where a subset of squares are barriers (but all non-barrier squares are connected). In the wake phase, the maze is updated by adding one barrier square and removing another (see Figure 6a, Appendix A.3). This means that the lower-level agent must update its strategy by replaying or simulating trajectories in the new maze.

In the wake phase the hippocampus receives 50 trajectories in the maze, before the meta-controller takes three actions in the sleep phase. The reward in the next wake phase is the performance of the lower-level maze solver agent on the same maze. The agent's inputs are the number of steps since the sleep phase began, the current accuracy of the world model, and the most recent action. The action space consists of three actions (see Figure 1b), closely analogous to those for the image classification task:

- **Action 0**: Update the world model from memories. This is a simple network for predicting the next state given a state and action, which is sufficient to allow the simulation of new episodes. (In a deterministic environment, the world model could simply cache the result of the most recent observation of a state-action pair.)

- **Action 1**: Train the maze solver on *stored* episodes from the model hippocampus (50 episodes oversampled to 200).

- **Action 2**: Train the maze solver on 200 *generated* episodes, in which the agent acts *within* the world model, i.e. epsilon-greedy rollouts are simulated.

In addition, the system learns which episodes are most valuable to replay or to simulate, similarly to the approach in the image task described above. Specifically, the Shapley values of a subset of the candidates (50 out of 576 possible start-goal combinations, in the case of generative replay) are obtained, using the world model to estimate the improvement to the agent. Then a value estimator predicts the utility of a particular start-goal pair. Unlike in the image case, this does not require a

real validation set, as the world model can be used to quantify the benefit. That is, the system can *simulate* how well a variant of the model trained on a subset of the training data would do.

### 3.2.1 Maze results

Figure 3a shows the learned schedule with which the meta-controller improves the maze solver; if only a few sequences are stored, it is more effective to learn the transition structure of the maze then simulate episodes than it is to learn from the limited data in the hippocampus. The meta-controller's 'success rate' of reaching the goal is higher than action selection and data selection baselines (Tables 3 and 4).

Even though the agent is tested only on the current maze, its learned strategy of training a generative model then learning from simulations resembles that of the image agent in the imbalanced case. This is because RL agents perform poorly with very limited training data, so learning from the hippocampus alone is not feasible. In addition, the world model in the maze task generates higher quality samples than the world model in the image case, so there is less of a trade-off associated with generative replay.

Start-goal pairs for which the shortest path was changed in the most recent maze update are higher value in both stages of generative replay (Figure 3c), but the relationship between shortest path length and value varies with learning (Figure 3d). We can also inspect the average estimated value of maze trajectories passing through each point in the maze. In some cases, locations nearby a recent change are more valuable at first, but learning saturates in later stages (Figure 3b), though these effects are not significant (Figure 3e).

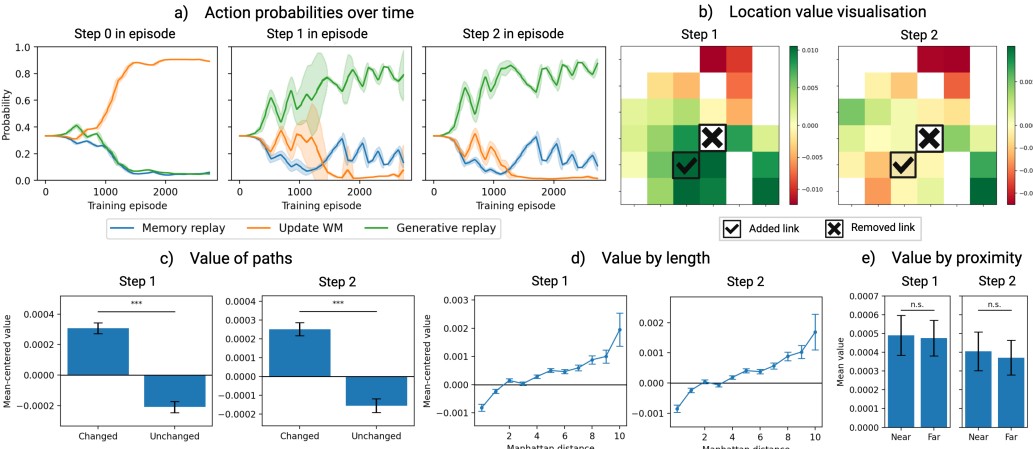

Figure 3: Maze simulations. a) Action probabilities for each position in the sleep sequence over the course of training. (In other words, step 0 is the first action chosen in the sleep phase, and so on.) The agent learns to update the world model then simulate new episodes with it. Shading indicates the SD across a sliding window of 100 episodes. b) Visualising the learned values of different locations in the maze. Each square shows the mean estimated value of all candidate episodes which pass through that point. The most recently added and removed squares are indicated. c) The mean-centered value of start-goal pairs for which the shortest path is changed vs. unchanged, for two stages of generative replay. Error bars give the SEM. d) The mean-centered value of start-goal pairs by their shortest path length, for two stages of generative replay. e) The mean value of locations near vs. far from maze changes.

### 3.3 Relational knowledge model

Biological intelligence excels at generalising from relatively few examples. In our framework, can the agent learn how to infer as much information as possible offline from a small number of online observations? We trained and tested the model on two relational inference tasks: spatial inference, and family tree inference, as in [37], although the approach is applicable to any scenario in which observations can be expressed as edges in a graph with an underlying structure. In the family tree

Table 3: Maze task results for the meta-controller's action selection compared to memory replay only baseline (with random data selection for both), averaged across ten incremental maze updates for three different random seeds.

| Approach | Success rate | SEM |
|---|---|---|
| Meta-controller | **0.822** | 0.007 |
| Memory replay | 0.161 | 0.005 |

Table 4: Maze task results for the data valuation method compared to baselines (all with meta-controller action selection), averaged across mazes and seeds as in Table 3.

| Approach | Success rate | SEM |
|---|---|---|
| Data valuation | **0.871** | 0.010 |
| Random selection | 0.822 | 0.007 |
| Longest paths | 0.736 | 0.013 |

inference task, the goal is to predict as many relationships as possible from a subset of family relationships (see Figure 4b), while in the spatial inference task, the goal is to predict the relative directions of locations in a 2D grid (i.e. whether each location is east / west / north / south of every other location).

In the wake phase the hippocampus receives 20 observed relationships, corresponding to a few of the edges of a true underlying graph. The agent interacts with a relational graph in each environment, iteratively updating its knowledge by choosing various meta-actions. In the previous simulations, the world models were used to generate new items in isolation with which to train a model to perform a given task. Here the model trained offline has a different function, inferring new facts to flesh out partial observations. Specifically, the world model is a graph autoencoder, with graph convolutional layers which allow it to learn shared patterns across graphs. Its decoder predicts missing edges in a graph, allowing the agent to infer new relationships. (Note that for training efficiency the model is cached, rather than retrained repeatedly.)

The meta-controller, a recurrent PPO agent as in the previous simulations, receives a three-dimensional observation vector consisting of the number of relationships learned so far, the number of edges in the current graph, and the accuracy of the world model. The action space consists of six discrete meta-actions:

- **Action 0**: Learn the task (i.e. add relationships to a list of known facts) from the model hippocampus. Here the task is trivial, but this could be some other operation on the knowledge, e.g. identifying the youngest members of the family, or finding the shortest path to a goal location.

- **Action 1**: Transfer (or 'consolidate') observed relationships in the model hippocampus into the knowledge graph.

- **Action 2**: Learn the task (i.e. add relationships to a list of known facts) from the knowledge graph.

- **Action 3**: Train the world model on all instances of graphs that have been built so far during meta-training.

- **Action 4**: Expand the knowledge graph using predictions from the world model. For example, if the world model has learned that a parent of a parent of X is a grandparent of X, applying the world model to the graph would add inferred grandparent edges to a family tree.

- **Action 5**: Do nothing. This avoids a penalty of 0.2 for all other actions, to incentivise efficient behaviour, and reflects the cognitive cost of taking the other actions.

The episode ends when either 10 meta-steps have been reached or a random stopping probability (0.05) triggers termination. At this point, the reward is computed based on the number of correct relationships inferred by the model compared to the true graph. The data valuation aspect is omitted in this simulation for simplicity.

### 3.3.1 Relational knowledge results

The meta-controller learns a schedule for offline actions to maximise online reward (Figure 4c and d). In the spatial task, the agent first learns from memories in the hippocampus, which is desirable because the episode can stop at any point, and this ensures at least some reward is obtained. Then the observed relationships are transferred (or 'consolidated') into the knowledge graph, as this is a prerequisite for relational inference to take place. After this, the world model is updated. (Note

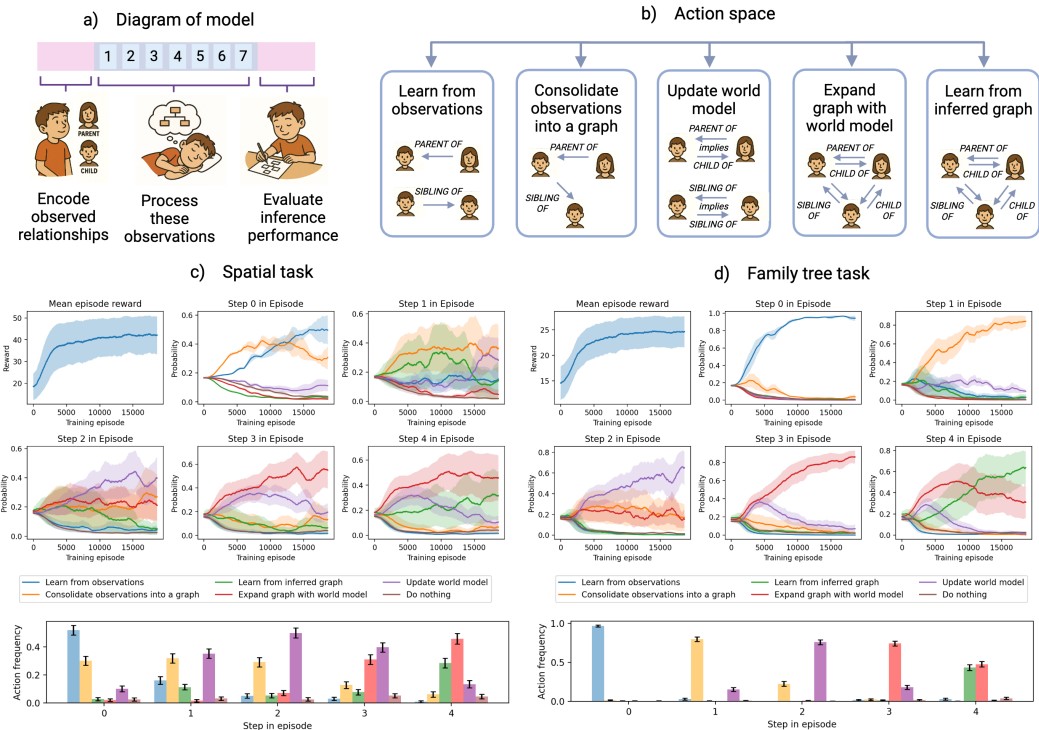

Figure 4: Relational knowledge simulations. a) Diagram of the model. At the start of each episode, twenty observations from a new graph are added to the hippocampus. The meta-controller learns to choose between several actions for expanding this knowledge, taking up to ten steps. The reward is the number of edges in the true graph which can be inferred by the end of the episode. b) The possible actions are: learning a given task from the observations in the model hippocampus, 'consolidating' observations from the hippocampus into a graph, updating a world model that captures relational structure, expanding the graph with new edges based on this world model, and learning a given task from the inferred graph. (Here the task is simply to build up a list of facts, but this could be some other operation on the knowledge, e.g. identifying the youngest members of the family, or finding the shortest path to a goal.) c) Above: Mean reward and action probabilities over the course of training for the spatial model, where the probabilities are plotted separately for each position in the sleep sequence. Shading indicates the SD across a sliding window of 100 episodes. Below: The final frequencies of actions chosen at each position, calculated across 20 episodes. Error bars give the SEM. Note that only the first five steps are shown. d) Likewise but for the family tree task.

that in this simple demonstration, the world model must be relearned in each episode, but one would expect this choice to be modulated by the accuracy of the world model if it were not reset.)

Next, the meta-controller uses the trained world model to infer additional relationships in the graph, e.g. inferring the fact that X is west of Z from the facts that X is west of Y and Y is west of Z. This action can be applied multiple times, iteratively expanding the graph. With each application of the graph convolutional autoencoder, links are predicted with a given probability, and those with a probability greater than 0.5 are added (see Figure 5, Appendix A.3 for an example of the iterative expansion of a family tree graph from a few edges). The chance of learning from the inferred graph ramps up later in the episode, as does the chance of doing nothing, as this avoids the small penalty for all other actions. The family tree task displays a similar pattern.

## 4 Discussion

We trained RL agents to perform metacognitive actions in an offline 'sleep' state in order to improve the 'awake' capabilities of a lower-level agent / model. The actions correspond to different processes

observed in biological offline learning, namely learning directly from memories, the training of predictive models, and the simulation of new events.

We found that the agents learned an effective curriculum for offline learning. In the maze navigation task, the agent learned to first update a world model capturing transition statistics, then simulate episodes with this model. The learned strategy depended on the demands of the task. In an image classification scenario requiring continual learning, the agent updated its generative model after receiving new observations, then learned from generated samples. Whilst these samples were lower quality, generative replay improved reward in the wake state by reducing catastrophic forgetting. But in the case that the agent observed all relevant categories in each wake state, this trade-off was no longer worthwhile, with the agent reverting to learning from real images in the model hippocampus. This could make predictions about how the brain regulates the replay of recent and remote examples to enable lifelong learning. The framework is flexible in that other metacognitive actions can easily be added, as demonstrated in the relational inference simulations, in which the agent learned to extract statistical patterns and then apply these to iteratively infer new facts.

In our simulations, the meta-controller also learns to choose *which* events to replay or simulate. For example, in the image task, easier examples (those closer to the mean of the class) were selected earlier in learning, whereas harder examples were selected later in learning. This was observed for both memory and generative replay. One might expect this to be true over short timescales (within a single block of sleep or rest), and over longer timescales while learning is gradually acquired. This lays the groundwork for making normative predictions about the optimal schedule of replays for learning, which could be compared to experimental neuroscience data. Previous work suggested that offline replay of memories is regulated by their noise level, so that noisy memories do not impair neocortical generalisation [38]. On the other hand, novelty is proposed to promote memory encoding [39]. Our approach could reconcile these different proposals about the relationship between schema congruence and replay by explaining how the optimal curriculum varies over the course of offline learning.

This work has several limitations, which reflect the fact that this is a very simplified model. Firstly, there are many variables that affect the behaviour of the data valuation component, so thorough tests of the sensitivity of the results to these parameters should be performed before we draw any general conclusions about which episodes are optimal to replay or simulate. Secondly, the data valuation aspect is computationally expensive, especially when fine-tuned in each stage of memory or generative replay. It would be more powerful for the data valuation network to learn to predict value based partly on information about the current progress of learning. Then the network could predict different values for the same items of data at different stages of training, without the need to fine-tune the model. Thirdly, a drawback of the current Shapley value-based data valuation mechanism in the image task is that it assumes a small validation set is available to the brain. The model could be refined to avoid this assumption, e.g. by using a subset of items in the hippocampus or generated data as a validation set (the latter is demonstrated in the maze task).

There are several other directions for future research. Here we only consider how offline processes are orchestrated, but an agent could also learn the optimal way to encode and retrieve memories. Such an agent might learn to only encode memories when it is worth the cognitive cost to do so (see also [40]). Similarly, one could integrate a mechanism for learning abstractions, e.g. 'options' in the maze navigation task. If the feedback in the wake state included a penalty for the cognitive cost of behaviour, so that solutions that combined a few simple components were advantageous, this might incentivise the meta-controller to develop 'primitives'.

In conclusion, we present a novel framework in which a meta-controller chooses between different metacognitive actions during 'sleep', inspired by the processes observed in resting brains. The meta-controller orchestrates offline learning to optimise 'wake' performance, and learns a curriculum for what to replay or simulate.

## Acknowledgements

Thanks to Rachel Swanson, Kris Jensen, Xiao Qin, Diksha Gupta, Mathias Sablé-Meyer, and Thomas Akam for helpful discussions.

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

# A  Appendix

Code for all simulations can be found at `https://github.com/ellie-as/rl-with-metacognitive-actions`.

Simulations were run on Linux virtual machines with NVIDIA A100 GPUs, and on MacOS with the MPS backend for GPU support.

Diagrams were created using BioRender.com, with some icons created using ChatGPT.

## A.1  Further model details

### A.1.1  Proximal policy optimisation

Actor–critic methods are a family of reinforcement learning algorithms which jointly learn both a policy (the 'actor') and a value function (the 'critic'), where the critic's value predictions guide the learning of the actor.

Proximal policy optimisation [25] is one such actor-critic algorithm, which adjusts the policy in small steps to avoid destabilising training.

In PPO, batches of experience collected with the current policy $\pi_{\theta_{\text{old}}}$ are used to update the parameters $\theta$ of the policy by maximising a clipped objective:

$$L^{\text{CLIP}}(\theta) = \mathbb{E}_t \Big[ \min\big(r_t(\theta)\,\hat{A}_t,\ \text{clip}\big(r_t(\theta), 1 - \varepsilon, 1 + \varepsilon\big)\,\hat{A}_t\big) \Big],$$

where:

- $r_t(\theta) = \dfrac{\pi_\theta(a_t \mid s_t)}{\pi_{\theta_{\text{old}}}(a_t \mid s_t)}$ is the ratio of the new and old action probabilities at time $t$.

- $\hat{A}_t$ is an estimate of the advantage, i.e. how much better the chosen action $a_t$ is compared to the average.

- $\varepsilon$ is a small clipping parameter (e.g., $0.2$) that limits how far $r_t$ can deviate from 1.

The clipping prevents large policy updates that could reduce performance.

To learn the right *sequences* of offline actions, we use a recurrent PPO network so the agent can condition its choices on the history of previous steps within the same 'sleep' phase. We use the default LSTM network (but with one hidden layer of 64 units) provided for a recurrent PPO agent in Stable-Baselines3 [28].

### A.1.2  Estimating value via Shapley values

To teach the agent which images or trajectories are most useful to replay, we use a small validation set and compute approximate Shapley values [15] for a subset of candidates. The Shapley value $\phi_i$ for example $i$ measures its average contribution to performance across all subsets of examples. Formally,

$$\phi_i = \sum_{S \subseteq N \setminus \{i\}} \frac{|S|!\,(|N| - |S| - 1)!}{|N|!} \big[v(S \cup \{i\}) - v(S)\big],$$

where $v(S)$ is the classifier accuracy (or maze solver performance) trained on subset $S$. The second term therefore measures how much adding example $i$ after subset $S$ increases performance. The first term gives the probability that, if you shuffle all $N$ examples into a random order, the set of items that appear before $i$ is $S$. This sum is therefore the expected marginal contribution of item $i$ when you add it into a randomly shuffled training set.

Because calculating this exactly is very expensive, Shapley values can be approximated by sampling random subsets of the candidates, with and without a particular pair. In the image experiments, we use standard data Shapley: for each training image, we estimate its value by sampling coalitions of images, training the classifier on each coalition with and without the target image, and averaging the

marginal accuracy gains. In the maze experiments, we value all candidate start–goal pairs in a single pass per permutation for efficiency. For each sampled ordering, we sequentially train on each pair and record the marginal performance gain it produces, then we average across all permutations as in standard data Shapley.

These estimated Shapley values for a subset of the candidates are used to train a value estimator model which quickly predicts value for every candidate. In the image task, the input to the value estimator is the image, whilst in the maze task, the input to the value estimator is the start-goal combination, represented as co-ordinates of the start concatenated with co-ordinates of the goal, and the distance between the start and goal.

### A.1.3   Selecting examples with maximal marginal relevance

To pick a set of items (real or generated) that are both informative and diverse, we use maximal marginal relevance [27], a common approach for information retrieval. We add examples one at a time to a set $S$, at each step scoring a candidate item $d$ by:

$$\text{MMR}(d) = \lambda \, \text{Relevance}(d) - (1 - \lambda) \max_{d' \in S} \text{Similarity}(d, d'),$$

where:

- $\text{Relevance}(d)$ is how valuable the example is predicted to be (from our value estimator),
- $\text{Similarity}(d, d')$ measures how similar two items are (e.g. cosine distance between the pixel vectors in the image task, or the sum of distances between the start locations and goal locations in the maze task),
- $\lambda \in [0, 1]$ balances relevance versus diversity. This is set to 0.9 by default in our simulations.

We then add the item with the highest MMR score, and repeat until the batch is full.

### A.1.4   Gaussian mixture model for image generation

To model generative replay in the image classification task, we use a Gaussian mixture model (GMM) as a simple generative model of images, implemented with scikit-learn [41]. This assumes that images from each class come from a mixture of $K$ Gaussian components. For each class $c$, the probability of an image $\mathbf{x}$ is:

$$p(\mathbf{x} \mid y = c) = \sum_{k=1}^{K} \pi_{c,k} \, \mathcal{N}\big(\mathbf{x} \mid \boldsymbol{\mu}_{c,k}, \, \boldsymbol{\Sigma}_{c,k}\big).$$

Here $\pi_{c,k}$ are mixing weights that sum to 1, and each Gaussian component has mean $\boldsymbol{\mu}_{c,k}$ and covariance $\boldsymbol{\Sigma}_{c,k}$. The parameters are found by the expectation–maximisation algorithm.

Images can then be generated by sampling from the learned distribution for a particular class.

### A.1.5   Deep Q-network maze solving agent

The DQN agent uses Stable-Baselines3's DQN with an MLP Q-network: a feed-forward network mapping the observation vector to action-values, with the default architecture of two fully connected hidden layers (64 units each) and ReLU activations, followed by a linear output over actions.

Training uses a learning rate of 1e-4, with epsilon fixed at 0.2 (higher than typical because of the frequent changes to the maze, which require significant exploration).

### A.2   Further neuroscience background

Recent memories are encoded in the hippocampus, which is capable of 'one-shot learning' of particular events. During sleep and rest, the brain conducts offline processing to better support future behaviour. In particular, replay is a phenomenon in which hippocampal neurons reactivate memories in 'fast-forward' [3]. In machine learning terms, hippocampal sequences are reactivated

to train other brain regions, with the neocortex thought to gradually learn statistical patterns across memories through replay [42, 43, 5, 44]. This view is supported by the fact that interventions which interfere with replay lead to increased forgetting [45], and replay after learning a task is correlated with subsequent performance [46]. (Note that much of this research relates to rodents performing spatial tasks like navigating to a reward in a maze, motivating our use of a maze as one scenario in the paper.)

A newer view is that replay may be generative rather than, or as well as, veridical. In rodents, 'replayed' sequences can join together paths that were experienced on separate occasions [6], cross regions that have been seen but not visited [47], and even diffuse throughout an open environment [48]. In human neuroimaging studies, 'replayed' sequences do not always correspond to real memories either [49]. In addition, conceptual 'building blocks' are thought to be learned offline which can then be flexibly recombined.

These findings have led to a body of research about what kinds of replay are observed when, and which events are replayed or simulated during rest. This is thought to depend on several factors. For example, consider the case of a gradually changing maze. Simulating trajectories offline without first updating the world model is detrimental if the world has changed, but updating the world model is costly if the world has not changed. So to optimise offline activity for future behaviour, there must be some mechanism for assessing which offline action to take based on the current cognitive state (e.g. whether recent experiences diverge from the world model's predictions).

In summary, during rest, the brain is thought to perform some combination of i) replaying real memories, ii) learning a general world model, and iii) simulating events with this world model. There is evidence for all of these processes, but there is little understanding of how they are regulated, and how this depends on the environment and/or current learning. The 'metacognitive actions' chosen in the paper are intended to reflect these observed offline processes in the neuroscience literature.

## A.3    Supplementary results

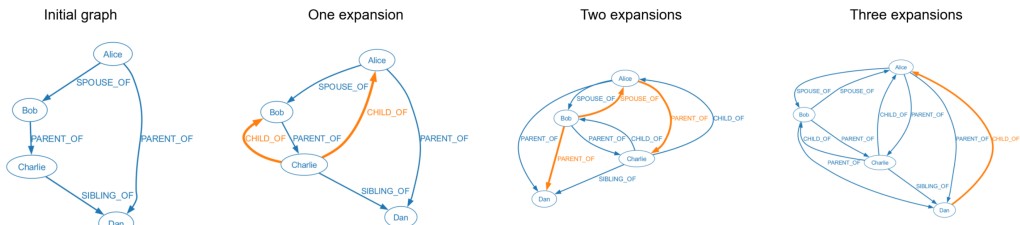

Figure 5: A toy example of iterative graph expansion. From left to right: The initial family tree graph is shown. When the graph autoencoder model for link prediction is applied to this graph, the links shown in orange, which have a probability greater than 0.5, are added. The graph autoencoder is then applied to this expanded graph.

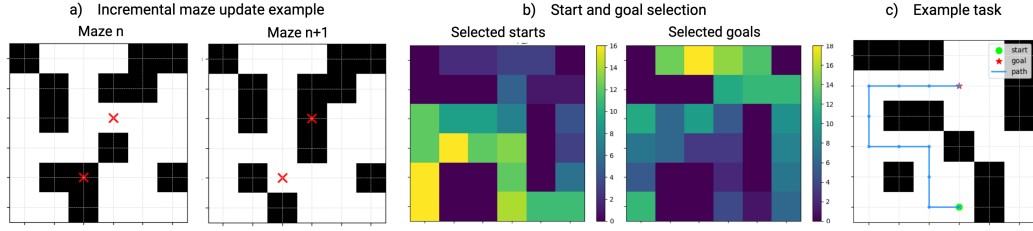

Figure 6: Additional maze visualisations. a) Incremental maze updates break a high centrality link and add another, whilst keeping the maze fully connected. b) Example heatmaps showing the frequency of selected start (left) and goal (right) locations. MMR is applied to select start-goal combinations which balance high value and high diversity. The selected start and goal locations may very different. c) An example task solved by the DQN agent.

