# OpenReview forum: "Modelling the control of offline processing with reinforcement learning"
_NeurIPS.cc/2025/Conference — NeurIPS 2025 poster_

### Official Review · Reviewer_9NGj · 2025-06-29

**Clarity:** 3
**Significance:** 3
**Originality:** 3
**Rating:** 5
**Confidence:** 4

**Summary:**

This paper proposes a computational framework to explain how the brain might effectively balance different offline learning strategies, namely consolidating direct experiences, abstracting a world model, and learning from imagined scenarios. The authors introduce a meta-reinforcement learning model where a meta-controller learns to guide the offline learning process of a lower-level agent. Specifically, during simulated sleep periods, this meta-controller selects from a set of metacognitive actions: training on real experiences, updating a world model, or training on experiences simulated from that model. The framework also learns to prioritize experiences by assigning them value. The paper's core contributions are to (1) provide a normative model that makes testable predictions about the content and timing of neural replay and offline processing, which the authors validate in three experiments, and (2) demonstrate how this method of constructing a bespoke learning curriculum can improve the overall performance of a reinforcement learning agent.

**Questions:**

- What is the performance of these models in comparison to heuristic baselines?
- How are metacognitive actions designed?
- For the maze experiment, are the start and end locations fixed?
- If the reward is the fraction of edges in the true graph which can be inferred by the end of the episode, why is the y-axis of 4(c) and 4(d) (top-left) showing values greater than 1? Is this actually showing a percentage.
- Is it a realistic assumption that the meta-controller always observes how accurate its world model is? What does this mean? Where does it get this information from?
- “In our simulations, the meta-controller also learns to choose which events to replay or simulate.”
    - Does the meta-controller do that, or the separate valuation network?

**Ethical Concerns:**

["NO or VERY MINOR ethics concerns only"]

**Final Justification:**

The updates to the results made by the authors in their rebuttal have addressed well my central critique of the paper, namely that one of the central claims initially lacked any supportive evidence in the quantitative results. Given that, I have decided to revise my score upwards accordingly.

**Limitations:**

Yes

**Quality:**

3

**Strengths And Weaknesses:**

Strengths:
- Generally speaking, the paper is clearly written and well organized. It is clear what the contribution of the paper is (the framework), how it works (in the model section), as well as the two key claims it makes regarding that contribution.
- The paper does provide an interesting framework that one could see validated in a range of other tasks (for instance, in an RLHF domain).
- As the paper points out, it also lays the groundwork for fruitful comparisons to experimental neuroscience data on the optimal scheduling of replays.
- This paper novelly employs meta-reinforcement learning in a neuroscience context when looking at the idea of metacognitive control.

Weaknesses:
- One of the central claims--that metacognitive actions and custom curriculum design improve performance--is not well-supported. To wit:
    - In Experiment 2, performance is only evaluated in the narrow domain of evaluating the data valuation component of the model, showing that this improves accuracy over random replay using both real and simulated examples.
    - In Experiment 3, no measure of performance (that is, the agent’s realized ability to efficiently navigate the maze) is reported at all.
    - In all three experiments, no baselines are evaluated showing how learning a policy for meta-cognitive actions would improve performance over a fixed-heuristic model like Dyna, Prioritized Sweeping, or its deep learning counterparts. These should be easy baselines to show in order to buttress this claim. Visualizing action policies is not enough.
- Fig. 3(c) is hard to interpret the values. They’re all too small to be seen in the annotation, but the colors are all different.
- The normative neuroscience predictions could be better elaborated upon.

---

> ### Author Rebuttal · Authors · 2025-07-31
>
> Thank you for your comments. We are glad you found the proposed framework interesting, and the application of meta-RL in the context of metacognitive control novel.
>
> We agree that comparison to baselines would strengthen the paper, so we have run additional simulations to show that metacognitive control of offline processing and custom curriculum design improve performance. (Due to restrictions on the rebuttal format, we summarise the results below, but we have prepared improved figures to add if the paper is accepted, as well as additional text. Results below give the mean across several repeats, but the improved figures have error bars.)
>
> **Metacontroller action selection baselines**
>
> In the image task, we wanted to compare the metacontroller’s adaptive strategy to i) a fixed memory replay strategy and ii) a fixed world model update + generative replay strategy. The table below shows that the two baselines each do well in only one setting, with the highest average accuracy across the two settings obtained by the metacontroller. Note that we have set the data valuation to be random to isolate the effect of the actions, and give results for Fashion-MNIST only:
>
> | Approach                | Accuracy&nbsp;(CL) | Accuracy&nbsp;(non‑CL) | Mean accuracy |
> |-------------------------|:------------------:|:----------------------:|:-------------:|
> | Metacontroller          | 0.36              | 0.41                   | 0.39          |
> | Memory replay only      | 0.14              | 0.41                   | 0.28          |
> | Generative replay only  | 0.34              | 0.30                   | 0.32          |
>
> In the maze task, we wanted to compare the metacontroller’s adaptive strategy to a fixed memory replay strategy (which performs very poorly as there are not enough real episodes to learn a good policy from, meaning that the metacontroller is better at generalising to new tasks). As above, we have set the data valuation to be random to isolate the effect of the actions:
>
> | Approach               |  Mean reward |
> |------------------------|:-----------:|
> | Metacontroller         | 0.90        |
> | Memory replay only     |  0.47        |
>
> **Data valuation baselines**
>
> We also tested new baselines for the data valuation component of the model. In the image task, we previously tested random selection as the only alternative to the learned data valuation network. To add to this, we tested training on the easiest and hardest images only (where the difficulty of an image was quantified as the fraction of an ensemble of scikit-learn classifiers that labelled the image correctly). Both perform worse than regulating the difficulty by the stage of learning (here we give results for just the non-CL setting with Fashion-MNIST):
>
> | Data‑valuation approach | Mean accuracy |
> |-------------------------|:-------------:|
> | Data valuation network  | 0.58          |
> | Random selection        | 0.41          |
> | Most challenging     | 0.44          |
> | Least challenging      | 0.34          |
>
> The fact that none of i) most challenging, ii) least challenging, or iii) random data alternatives do better provides support for the claim that the system is choosing the optimal examples for its current stage of learning, i.e. that it is learning a curriculum from less challenging to more challenging data.
>
> In the maze task, the only previous baseline was random selection, so we tried two baselines that correspond to simple heuristics for which episodes are most useful to the agent: i) training on the longest trajectories, and ii) training on only paths that visit the new link. Both performed worse than the data valuation network, suggesting the data valuation network learns something more sophisticated than these simple heuristics.
>
> | Data‑valuation approach | Mean accuracy |
> |-------------------------|:-------------:|
> | Data valuation network  | 0.94          |
> | Random selection        | 0.90          |
> | Longest paths           | 0.85          |
> | Paths through new link  | 0.87          |
>
> We hope this addresses the first question, and the first weakness identified in your review.
>
> Answers to your other questions are as follows.
>
> * *How are metacognitive actions designed?* The metacognitive actions are chosen to reflect a highly simplified version of metacognitive processes discussed in the neuroscience literature. See the response to reviewer mBgr for ore explanation of evidence for these actions.
> * *For the maze experiment, are the start and end locations fixed?* No, the start and end locations can be anywhere in the maze.
> * *If the reward is the fraction of edges in the true graph which can be inferred by the end of the episode, why is the y-axis of 4(c) and 4(d) (top-left) showing values greater than 1? Is this actually showing a percentage?* Apologies, this shows the number of edges, not the fraction of edges. We have corrected this error.
> * *Is it a realistic assumption that the meta-controller always observes how accurate its world model is? What does this mean? Where does it get this information from?* We assume a small test set exists, which could be a subset of hippocampal memories. If this subset is not used to update the world model, it can instead be used as a test set to evaluate it, by tracking the prediction error of the world model on the test set.
> * *“In our simulations, the meta-controller also learns to choose which events to replay or simulate.” Does the meta-controller do that, or the separate valuation network?* We were using meta-controller to refer to the combined system, but you are correct that we should be more precise, as this claim could be confused with the recurrent PPO agent itself choosing events to replay or simulate. We have clarified this.
>
> In addition, you pointed out that ‘In Experiment 3, no measure of performance (that is, the agent’s realized ability to efficiently navigate the maze) is reported at all’. We have added a plot to show that the agent’s mean reward per episode in the awake state increases over time as the meta-controller learns, which shows that once the meta-controller has learned to sequence its meta-actions correctly, reward is near optimal (with optimal performance plotted for comparison). We have also shown trajectories for a random subset of (start, goal) pairs to illustrate this qualitatively.
>
> Thanks again for your feedback- we hope you’ll consider updating your evaluation if we have addressed your concerns. If accepted, the additional results will obviously be included in the final version, as will the updated code to replicate them.

---

> > ### Comment · Reviewer_9NGj · 2025-08-02
> >
> > Thank you for responding to all of my points in detail. Before I make my final summary, can you also include in the tables the values corresponding to the error bars in the improved figures for the results you have given? This will help to give the fullest picture of your rebuttal, in spite of the constraints imposed by the OpenReview format.

---

> > > ### Author Response · Authors · 2025-08-03
> > >
> > > Thanks for your response.
> > >
> > > Here are the tables with the SEM added.
> > >
> > > Metacontroller action selection baselines:
> > >
> > > Image task:
> > >
> > > | Approach               | Accuracy&nbsp;(CL) | Accuracy&nbsp;(non-CL) | **Mean accuracy** | **SEM** |
> > > |------------------------|-------------------:|-----------------------:|------------------:|:------:|
> > > | Metacontroller     | 0.36 | 0.41 | 0.39 | 0.02 |
> > > | Memory replay only     | 0.14 | 0.41 | 0.28 | 0.03 |
> > > | Generative replay only | 0.34 | 0.30 | 0.32 | 0.02 |
> > >
> > > Maze task:
> > >
> > > | Approach           | **Mean reward** | **SEM** |
> > > |--------------------|---------------:|:------:|
> > > | Metacontroller | 0.90 | 0.02 |
> > > | Memory replay only | 0.47 | 0.08 |
> > >
> > >  Data valuation baselines:
> > >
> > > Image task:
> > >
> > > | Data-valuation approach | **Mean accuracy** | **SEM** |
> > > |-------------------------|------------------:|:------:|
> > > | Data valuation network | 0.58 | 0.05 |
> > > | Random selection        | 0.41 | 0.03 |
> > > | Most challenging        | 0.44 | 0.03 |
> > > | Least challenging       | 0.34 | 0.06 |
> > >
> > > Maze task:
> > >
> > > | Data-valuation approach | **Mean accuracy** | **SEM** |
> > > |-------------------------|------------------:|:------:|
> > > | Data valuation network| 0.94 | 0.01 |
> > > | Random selection        | 0.90 | 0.02 |
> > > | Longest paths           | 0.85 | 0.02 |
> > > | Paths through new link  | 0.87 | 0.03 |

---

> > > > ### Comment · Reviewer_9NGj · 2025-08-05
> > > >
> > > > Thank you very much for including these in your table. This shows a clear advantage of the metacontroller and the data valuation network over the other strategies.

---

### Official Review · Reviewer_1iZc · 2025-06-30

**Clarity:** 2
**Significance:** 2
**Originality:** 3
**Rating:** 4
**Confidence:** 3

**Summary:**

This paper prpopses a new framework to better use the acquired experiences for training. Specifically, a meta controller is used to decide whether the agent should choose from 1) training the model using real data, 2) build a world model from the data and 3) train the model using data generated by the world model. The method addresses some issues with previous methods such as the reliance on human judgement on the data augmentation/curriculum design process, and the low quality of synthetic data. Empirical study shows the method can effectively learn an adaptive curriculum for offline learning.

**Questions:**

My concerns listed in "weaknesses" are also my questions to the author(s). I will consider updating my evaluation if my concerns are addressed.

**Ethical Concerns:**

["NO or VERY MINOR ethics concerns only"]

**Final Justification:**

I raised my score as the authors addressed my concerns.

**Limitations:**

The authors adequately addressed the limitations and potential negative societal impact of their work.

**Quality:**

2

**Strengths And Weaknesses:**

Strengths:

1. The proposed method is novel, it provides a new view for curriculum design or data augmentation.

2. The idea is conceptually sound and easy to follow. The method is well described.



Weaknesses:

1. This paper lacks comparative study. While the existing experiments show that the framework can learn an effective curriculum for offline learning, comparisons to previous image classification, navigation and relational inference algorithms are needed to better demonstrate the framework's performance and potential advantages over these related work. For example, the author(s) may want to compare the classification accuracy with traditional image classifiers in the image task or compare the cumulative reward / success rate with offline RL algorithms in the navigation task.

2. Some ablation studies are needed to better understand the effect of each component in the proposed framework. An example to do this is to try (world model constructuion + learn from data generated only) to see the importance of training on real data. Another example is sample training data with some simple distribution instead of a shapley value estimator to show the effect of data valuation.

---

> ### Author Rebuttal · Authors · 2025-07-31
>
> Thank you for your comments. We are glad you found the general idea of the paper novel and conceptually sound.
>
> We agree that the paper would benefit from more comparison and ablation studies, so we have run additional simulations to show that metacognitive control of offline processing and custom curriculum design improve performance compared to sensible baselines. (Due to restrictions on the rebuttal format, we summarise the results below, but we have prepared improved figures to add if the paper is accepted, as well as additional text. Results below give the mean across several repeats, but the improved figures have error bars.)
>
> **Metacontroller action selection baselines**
>
> In the image task, we wanted to compare the metacontroller’s adaptive strategy to i) a fixed memory replay strategy and ii) a fixed world model update + generative replay strategy. The table below shows that the two baselines each do well in only one setting, with the highest average accuracy across the two settings obtained by the metacontroller. Note that we have set the data valuation to be random to isolate the effect of the actions, and give results for Fashion-MNIST only:
>
> | Approach                | Accuracy&nbsp;(CL) | Accuracy&nbsp;(non‑CL) | Mean accuracy |
> |-------------------------|:------------------:|:----------------------:|:-------------:|
> | Metacontroller          | 0.36              | 0.41                   | 0.39          |
> | Memory replay only      | 0.14              | 0.41                   | 0.28          |
> | Generative replay only  | 0.34              | 0.30                   | 0.32          |
>
> In the maze task, we wanted to compare the metacontroller’s adaptive strategy to a fixed memory replay strategy (which performs very poorly as there are not enough real episodes to learn a good policy from, meaning that the metacontroller is better at generalising to new tasks). As above, we have set the data valuation to be random to isolate the effect of the actions:
>
> | Approach               |  Mean reward |
> |------------------------|:-----------:|
> | Metacontroller         | 0.90        |
> | Memory replay only     |  0.47        |
>
> **Data valuation baselines**
>
> We also tested new baselines for the data valuation component of the model. In the image task, we previously tested random selection as the only alternative to the learned data valuation network. To add to this, we tested training on the easiest and hardest images only (where the difficulty of an image was quantified as the fraction of an ensemble of scikit-learn classifiers that labelled the image correctly). Both perform worse than regulating the difficulty by the stage of learning (here we give results for just the non-CL setting with Fashion-MNIST):
>
> | Data‑valuation approach | Mean accuracy |
> |-------------------------|:-------------:|
> | Data valuation network  | 0.58          |
> | Random selection        | 0.41          |
> | Most challenging     | 0.44          |
> | Least challenging      | 0.34          |
>
> The fact that none of i) most challenging, ii) least challenging, or iii) random data alternatives do better provides support for the claim that the system is choosing the optimal examples for its current stage of learning, i.e. that it is learning a curriculum from less challenging to more challenging data.
>
> In the maze task, the only previous baseline was random selection, so we tried two baselines that correspond to simple heuristics for which episodes are most useful to the agent: i) training on the longest trajectories, and ii) training on only paths that visit the new link. Both performed worse than the data valuation network, suggesting the data valuation networks learns something more sophisticated than these simple heuristics.
>
> | Data‑valuation approach | Mean accuracy |
> |-------------------------|:-------------:|
> | Data valuation network  | 0.94          |
> | Random selection        | 0.90          |
> | Longest paths           | 0.85          |
> | Paths through new link  | 0.87          |
>
> Thanks again for your feedback- we hope you’ll consider updating your evaluation if we have addressed your concerns. If accepted, the additional results will obviously be included in the final version, as will the updated code to replicate them.

---

> > ### Comment · Reviewer_1iZc · 2025-08-07
> >
> > Thank you for the response, the additional results and explanations make this paper clearer.

---

### Official Review · Reviewer_mBgr · 2025-07-02

**Clarity:** 1
**Significance:** 1
**Originality:** 1
**Rating:** 2
**Confidence:** 3

**Summary:**

The paper proposes an RL agent operating in two states, “sleep” and “awake,” inspired by work in cognitive science. Learning occurs only in the “sleep” state, and data collection occurs in the “awake” state.

**Questions:**

1. What is the proposed approach in formal mathematical notation?
2. What is the hippocampus model in the experiments?
3. What is the world model in the experiments?
4. Why were these particular experimental setups chosen?

The main drawback of the work is the lack of a general picture after reading the paper due to poorly constructed narration and positioning. I believe that the work requires significant revision in this part and is not ready for publication at the current stage.

**Ethical Concerns:**

["NO or VERY MINOR ethics concerns only"]

**Final Justification:**

The paper suffers from a lack of a clear formal description of the proposed approach and technical details. To eliminate this shortcoming, a significant revision of the paper is required. In this regard, I believe that in its current form the paper cannot be presented at such a top-tier conference as NeurIPS.

**Limitations:**

The limitations are described in sufficient detail.

**Paper Formatting Concerns:**

I did not notice any major formatting issues.

**Quality:**

2

**Strengths And Weaknesses:**

**Strengths:**

1. A large number of experiments, which nevertheless do not add up to a general picture.

**Weaknesses:**

1. The paper is poorly constructed at the narrative level, making it difficult to draw conclusions about the significance and originality of the work.

**Quality**

The paper presents a large number of experiments, but they do not fit into the overall picture and, due to the lack of a clearly formulated contributions, cannot confirm them.

**Clarity**

The work is very difficult to read and perceive, the narrative jumps between sections without linking the individual parts together.
1. There is no clear indication of the contribution of the paper in the introduction.
2. There is no clear formal description of the approach and problem statement, including through formulas and algorithms. Figure 1 gives only a general idea of ​​​​the method.
3. There is no motivation why these particular experimental setups were chosen.
4. The part from the Discussion section describing the limitations of the work would be better allocated to a separate section.

**Significance and Originality**

It is really hard to evaluate these aspects of the work because on the one hand there is no clear, well-articulated contribution of the work, on the other hand there is no clear positioning of the work relative to others in the Previous work section. Authors should clearly position themselves relative to the works mentioned.

However, I should note that I am not an expert in neuroscience and cognitive science and may be missing important results for these areas.

---

> ### Author Rebuttal · Authors · 2025-07-31
>
> Thank you for your comments. We’re sorry that the paper was unclear to those without a neuroscience background, and hope we can clarify the ‘general picture’ below.
>
> Here is a revised framing of the paper with some more neuroscience context:
> * Recent memories are encoded in the hippocampus, which is capable of ‘one-shot learning’ of particular events; as well as capturing our recent experiences, it is often thought of as a store of training data for other networks in the brain.
> * During sleep and rest, a lot is happening to update neural networks to better support future behaviour. In particular, replay is a phenomenon in which hippocampal neurons reactivate memories in ‘fast-forward’ (Foster, 2017). In ML terms, data are reactivated to train other brain regions, with the neocortex thought to gradually learn statistical patterns across memories through replay (Marr, 1970, 1971; McClelland et al., 1995; Teyler & DiScenna, 1986). Interventions that interfere with replay lead to increased forgetting (Girardeau et al., 2009), and replay after learning a task is correlated with subsequent performance (Peigneux et al., 2004). (Note that much of this research relates to rodents performing spatial tasks like navigating to a reward in a maze, motivating our use of a maze as one scenario in the paper.)
> * A newer view is that replay may be generative rather than, or as well as, veridical. In rodents, ‘replayed’ sequences can join together paths that were experienced on separate occasions (Gupta et al., 2010), cross regions that have been seen but not visited (Olafsdottir et al., 2015; Pfeiffer & Foster, 2015), and even ‘diffuse’ throughout an open environment (Stella et al., 2019). In human neuroimaging studies, ‘replayed’ sequences do not always correspond to real memories either (Liu et al., 2019). In addition, conceptual 'building blocks' are thought to be learned offline which can then be flexibly recombined.
> * These findings have led to a body of research about what kinds of replay are observed when, and which events are replayed / simulated during rest. This is thought to depend on the situation. To illustrate this, consider the case of adding and removing links gradually in a maze. One might expect that simulating trajectories offline without first updating the world model would be detrimental to learning. But updating the world model is costly to do when the world has not changed. So if the brain is optimising its offline activity for future behaviour, there must be some mechanism for assessing which offline action to take based on the current cognitive state (e.g. whether recent experiences diverge from the world model’s predictions).
> * So, during rest, the brain is doing some combination of 1) replaying real memories, 2) learning a general world model, 3) simulating events with this world model, and 4) learning specific tasks. There is evidence for all of these processes, but there is little understanding of how they are regulated, and how this depends on the environment and/or the animal’s current learning. The ‘metacognitive actions’ chosen in the paper are intended to reflect these observed offline processes in the neuroscience literature.
> * We want to develop an approach to understand what sequence of offline processing is optimal, and which events should be replayed / simulated to optimise future behaviour. This could make predictions about the pattern on offline activity that could be tested in future experiments.
> * Whilst the training data in modern ML tends to be carefully curated, e.g. by balancing and interleaving different categories, brains excel at data-efficient learning in a changing environment, in which generalisable knowledge must be extracted from limited, noisy experience. The offline processing of memories and beliefs is thought to support this, so we hope these mechanisms are of wider interest to the ML community, and could lead to brain-inspired innovations.
>
> I hope this provides some more insight into why the orchestration of offline processing is an important unsolved problem in neuroscience, motivating our study of whether a meta-controller could learn when to perform different ‘metacognitive actions’. We have updated our draft to make the paper assume less prior knowledge.
>
> In response to your specific questions above:
> 1. The approach is described more mathematically in the supplementary information of the initial version, but we have extended this to provide more details. As a brief summary:
>
> For each sleep/awake cycle $n = 1,\\ldots,N$:
>
> *Awake phase:* Agent $\\theta_n$ interacts with the environment for $K$ steps (i.e. ${n\_{episodes}}$), encoding memories $H\_n = \{(s_k,a_k,r_k)\}\_{k=1}^{K} $ in the model hippocampus and recording an overall reward for the awake phase $R\_n = \tfrac{1}{n\_{episodes}}\\sum\_{k=1}^{K} r\_k$.
>
> *Sleep phase:* A meta-controller takes $T$ offline actions $a_{n,0:T-1}$, corresponding to the kinds of offline processing observed in the neuroscience literature described above.
>
> *Objective:* The recurrent PPO policy $\\pi_{\\phi}(a \\mid x)$ updates its parameters to maximise the expected cumulative return $\mathbb{E}\_{\pi\_{\phi}}\left[\sum\_{n=1}^{N} R_n\right]$.
>
> In the maze task, the reward $R_n$ is the mean reward per episode of the lower-level agent in the awake state, i.e. in the real maze environment. In the image task the reward is the mean classification accuracy for new images encountered in the awake state.
>
> 2. The hippocampus in our simulations is simply a list of episodes the agent has experienced in the awake state. However in reality we would envisage the hippocampus to be a network capable of one-shot learning of sequential experiences, e.g. an asymmetric modern Hopfield network (see Millidge et al., 2022).
> 3. The world model is a model that can simulate experience with which to train the model for the task. In the image case, the world model is the image generator. In the maze case, the world model is the transition model. However the ideas in this paper can be applied to more complex kinds of world model, e.g. consider an agent learning to play a video game from simulations with a world like PlaNet (Hafner et al., 2019) or DreamerV2 (Hafner et al., 2021). In a setting in which the video game environment changes over time, encoding and replay of new experiences to update the world model need to be balanced with learning from the world model, and our approach would be applicable in this context.
> 4. The image case is chosen because it is a classic setting for considering learning from generated data, and because it enables the data valuation network’s selections to be easily visualised. The maze case is chosen because a lot of neuroscience relates to rodents performing spatial tasks like navigating to a reward in a maze, and a maze is a natural task in which to consider changes to an environment over time (by adding or removing links). However, many kinds of low-level agent and environment can work within this framework. (If accepted the final version of the code will demonstrate this.)
>
> Thanks again for your feedback - we hope you’ll consider updating your evaluation if we have addressed your concerns.
>
> **References:**
>
> * Foster, D. J. (2017). Replay comes of age. Annual Review of Neuroscience, 40, 581–602.
> * Marr, D. (1970). A theory for cerebral neocortex. Proceedings of the Royal Society of London. Series B: Biological Sciences, 176(1043), 161–234.
> * Marr, D. (1971). Simple memory: A theory for archicortex. Philosophical Transactions of the Royal Society of London. B, Biological Sciences, 262(841), 23–81.
> * McClelland, J. L., McNaughton, B. L., & O’Reilly, R. C. (1995). Why there are complementary learning systems in the hippocampus and neocortex: Insights from the successes and failures of connectionist models of learning and memory. Psychological Review, 102(3), 419–457.
> * Teyler, T. J., & DiScenna, P. (1986). The hippocampal memory indexing theory. Behavioral Neuroscience, 100(2), 147–154.
> * Girardeau, G., Benchenane, K., Wiener, S. I., Buzsáki, G., & Zugaro, M. B. (2009). Selective suppression of hippocampal ripples impairs spatial memory. Nature Neuroscience, 12(10), 1222–1223.
> * Peigneux, P., Laureys, S., Fuchs, S., Collette, F., Phillips, C., Aerts, J., … Maquet, P. (2004). Are spatial memories strengthened in the human hippocampus during slow wave sleep? Neuron, 44(3), 535–545.
> * Gupta, A. S., van der Meer, M. A. A., Touretzky, D. S., & Redish, A. D. (2010). Hippocampal replay is not a simple function of experience. Neuron, 65(5), 695–705.
> * Ólafsdóttir, H. F., Barry, C., Saleem, A. B., Hassabis, D., & Spiers, H. J. (2015). Hippocampal place cells construct reward related sequences through unexplored space. eLife, 4, e06063.
> * Pfeiffer, B. E., & Foster, D. J. (2015). Autoassociative dynamics in the generation of sequences of hippocampal place cells. Science, 349(6244), 180–183.
> * Stella, F., Baracskay, P., O’Neill, J., & Csicsvari, J. (2019). Hippocampal reactivation of random trajectories resembling Brownian diffusion. Neuron, 102(2), 450–461.e7.
> * Liu, Y., Dolan, R. J., Kurth-Nelson, Z., & Behrens, T. E. J. (2019). Human replay spontaneously reorganizes experience. Cell, 178(3), 640–652.e14.
> * Hafner, D., Lillicrap, T., Fischer, I., Villegas, R., Ha, D., Lee, H., & Davidson, J. (2019). Learning latent dynamics for planning from pixels. arXiv preprint arXiv:1811.04551.
> * Hafner, D., Lillicrap, T., Norouzi, M., & Ba, J. (2021). Mastering Atari with discrete world models. arXiv preprint arXiv:2010.02193.

---

> > ### Comment · Reviewer_mBgr · 2025-08-08
> > **Answer to the authors**
> >
> > Thanks to the authors for the response and explanations, but my doubts were not dispelled. All technical information necessary for understanding the paper must be presented in the main text. Also, the framework should be clearly described by a general mathematical formalism. The specific implementation of certain functions may differ depending on the application, but not fall outside the general pipeline. If the authors refer to Appendix A2, it does not contain a general description of the entire framework, but only its individual modules that are not built into a common pipeline. I recommend that the authors present the formal and technical side of the work more clearly.
> >
> > After the new description and explanations, new questions arose that the authors should take into account when revising the paper:
> > 1. What does the \pi_{\theta} strategy refer to as an agent or a meta-controller?
> > 2. PPO is an on-policy algorithm, how can the data obtained using the world model be used for training?
> >
> > Considering all of the above, I leave my initial assessment.

---

### Decision · Program_Chairs · 2025-09-17

**Decision:**

Accept (poster)

**Comment:**

The paper presents a framework in which an agent learns to control states of learning and how to balance inputs.  Although one reviewer pointed out several weaknesses as the reason for lower scores, the authors seemed to have fully addressed these weaknesses and can easily integrate these findings into the paper.
The approach is novel, sound and the paper is well written, with some expected interesting impact.